# Anti-Müllerian Hormone Inhibits FSH-Induced Cumulus Oocyte Complex In Vitro Maturation and Cumulus Expansion in Mice

**DOI:** 10.3390/ani12091209

**Published:** 2022-05-07

**Authors:** Xue Yu, Zan Li, Xinzhe Zhao, Liping Hua, Shuanghang Liu, Changjiu He, Liguo Yang, John S. Davis, Aixin Liang

**Affiliations:** 1Key Laboratory of Agricultural Animal Genetics, Breeding and Reproduction of Ministry of Education, College of Animal Science and Technology, Huazhong Agricultural University, Wuhan 430070, China; yuxuefish@163.com (X.Y.); lzan2022@163.com (Z.L.); who_is_xinzhe.zhao@webmail.hzau.edu.cn (X.Z.); 1737884213@webmail.hzau.edu.cn (L.H.); shuanghang_liu@webmail.hzau.edu.cn (S.L.); chungjoe@mail.hzau.edu.cn (C.H.); yangliguo2006@foxmail.com (L.Y.); 2Shandong Provincial Key Laboratory of Biophysics, Institute of Biophysics, Dezhou University, Dezhou 253023, China; 3National Center for International Research on Animal Genetics, Breeding and Reproduction (NCIRAGBR), Wuhan 430070, China; 4Olson Center for Women’s Health, Department of Obstetrics and Gynecology, University of Nebraska Medical Center, Omaha, NE 68198, USA; jsdavis@unmc.edu

**Keywords:** anti-müllerian hormone, follicle-stimulating hormone, oocyte, maturation, mice

## Abstract

**Simple Summary:**

Anti-Müllerian hormone (AMH) is a homodimeric glycoprotein composed of two identical subunits, which inhibits the recruitment of primordial follicles and the development of antral follicles in females. Anti-Müllerian hormone can be used as a diagnostic and prognostic marker for ovarian reserve, superovulation, embryo quality, and conception rate. However, few studies have focused on the effect of AMH on oocyte maturation. In the present study, we found Anti-Müllerian hormone has no effect on the nuclear maturation and cumulus expansion of cumulus oocyte complexes (COCs), whereas it has an inhibitory effect on follicle-stimulating hormone (FSH)-stimulated COCs nuclear maturation and cumulus expansion. These findings expand our knowledge of the functional role of AMH in modulating folliculogenesis.

**Abstract:**

Anti-Müllerian hormone (AMH) is secreted by the ovaries of female animals and exerts its biological effects through the type II receptor (AMHR2). AMH regulates follicular growth by inhibiting the recruitment of primordial follicles and reducing the sensitivity of antral follicles to FSH. Despite the considerable research on the actions of AMH in granulosa cells, the effect of AMH on the in vitro maturation of oocytes remains largely unknown. In the current study, we showed that AMH is only expressed in cumulus cells, while AMHR2 is produced in both cumulus cells and oocytes. AMH had no significant effect on COCs nuclear maturation, whereas it inhibited the stimulatory effects of FSH on COCs maturation and cumulus expansion. Moreover, AMH treatment effectively inhibited the positive effect of FSH on the mRNA expressions of Hyaluronan synthase 2 (*Has2*), Pentraxin 3 (*Ptx3*), and TNF-alpha-induced protein 6 (*Tnfaip 6*) genes in COCs. In addition, AMH significantly decreased the FSH-stimulated progesterone production, but did not change estradiol levels. Taken together, our results suggest that AMH may inhibit the effects of FSH-induced COCs in vitro maturation and cumulus expansion. These findings increase our knowledge of the functional role of AMH in regulating folliculogenesis.

## 1. Introduction

Anti-Müllerian hormone (AMH), also known as Müllerian inhibiting substance (MIS), is a dimeric glycoprotein and a member of the transforming growth factor β (TGF-β) superfamily [1,2]. AMH was first known for its regulatory role on müllerian ducts in male sexual differentiation [3,4]. More recently, AMH has been widely studied because of its potential clinical utility. In women, AMH can be used as a diagnostic and prognostic marker for ovarian reserve [5,6], polycystic ovary syndrome (PCOS) [7,8], implantation potential [9], primary ovarian insufficiency (POI) [10,11], and granulosa cell tumors [12,13]. In livestock, AMH is a predictive marker in herd longevity [14] and fertility including response to superovulation [15,16], outcome of in vitro embryos [16,17,18], and conception rate [19].

The clinical application of the studies cited above mostly depends on the expression pattern and biological function of AMH in the ovary, even though little is known about the factors that regulate AMH expression. In females, AMH has been reported to be highly expressed in granulosa cells of preantral and small antral follicles rather than primordial or atretic follicles [20,21], indicating AMH may play a crucial role in folliculogenesis. The function of AMH has been revealed through studies of AMH transgenic mice [22] and AMH-deficient mice [23]. Despite the lack of an obvious ovarian phenotype in AMH-deficient females, studies demonstrated that AMH could be involved in inhibiting primordial follicle recruitment [24]. These findings were subsequently confirmed by using an in vitro follicle culture system, showing that AMH inhibits the growth of preantral and antral follicles through regulating the sensitivity of FSH [25]. The AMH-induced inhibitory action on follicle growth was mainly the results of reduced granulosa cell proliferation, and decreased aromatase activity and estradiol production [26,27]. Additionally, AMH inhibits follicle activation in response to insulin in ovarian cortical fragments from bovine fetal ovaries in late gestation [28].

Besides the expression pattern mentioned above, AMH is expressed in cumulus cells of large and pre-ovulatory follicles [29]. Its specific expression pattern suggests that AMH, in the form of autocrine or paracrine, may regulate the maturation of the oocyte. However, there are few available studies and the views are contrary regarding the regulatory effect of AMH on oocytes. Takahashi et al. [30] observed that AMH inhibits the development of oocyte meiosis in rats, whereas another study indicated that AMH has no effect on oocyte meiosis [31]. Recently, Zhang et al. [32] revealed that AMH has no effect on in vitro COCs maturation rate in mice, but can improve the blastocyst rate. The reasons for different observations are unknown, but likely are due to differences in species and in vitro culture systems. The objective of the current study is to characterize the expression of AMH and its specific receptor (AMHR2) particularly in oocytes, and elucidate the regulatory effects of AMH on oocyte maturation, cumulus expansion, and steroidogenesis.

## 2. Materials and Methods

### 2.1. Ethics Statement

All animal experiments in this study were approved by the Scientific Ethic Committee of Huazhong Agricultural University (HZAUMO-2017-052) and were performed in accordance with the Guidelines for the Care and Use of Laboratory Animals of the Research Ethics Committee, Huazhong Agricultural University.

### 2.2. Animals

Immature female Kunming mice, aged 21 days old, were purchased from Hubei Disease Control and Prevention Center (Wuhan, China). Animals were housed in air-conditioned room at a constant temperature of 25 ± 2 °C with 12 h light/dark cycles, provided ad libitum with water and food. In order to obtain more immature oocytes at the germinal vesicle (GV) phase, the female mice were primed with an intraperitoneal injection of 7.5 IU Pregnant Mare Serum Gonadotropin (PMSG; Sansheng Pharmaceutical Corporation, Ningbo, China) and then sacrificed 44 h later by cervical dislocation.

### 2.3. Collection and In Vitro Maturation of Cumulus Oocyte Complexes (COCs)

Ovaries were collected from PMSG-treated mice, and follicles were pierced with a sterile 30 gauge needle to allow the COCs flow into the culture medium drop which contained α-MEM (Gibco, New York, NY, USA) supplemented with 5% FBS (R&D Systems, Minnesota, MN, USA), 20 ng/mL EGF (PeProtech, Cranbury, NJ, USA), 75 μg/mL penicillin (Gibco, New York, NY, USA), 50 μg/mL streptomycin (Gibco, New York, NY, USA), 20 mM sodium pyruvate (Sigma-Aldrich, St. Louis, MO, USA), and 3 mg/mL BSA (Sigma-Aldrich, St. Louis, MO, USA). Control group (α-MEM cell culture medium) and three experimental groups [α-MEM with 100 ng/mL rhAMH (R&D Systems, Minnesota, MN, USA), α-MEM with 100 ng/mL rmFSH (R&D Systems, Minnesota, MN, USA), α-MEM with 100 ng/mL rhAMH plus 100 ng/mL rmFSH] were placed in a small dish, and three droplets were prepared for each group. All droplets were covered with paraffin oil (Sigma-Aldrich, St. Louis, MO, USA) and incubated under 37 °C in a humidified incubator containing 5% CO_2_ for 16 h.

### 2.4. RNA Isolation and Quantitative Reverse-Transcription PCR Analysis

Total RNA in COCs, cumulus cells, and cumulus-free oocytes was extracted according to RNeasy Micro Kit (QIAGEN, Dusseldorf, Germany) instructions. cDNA was synthesized from 1 μg RNA of each sample by QuantiTect Reverse Transcription Kit (QIAGEN, Dusseldorf, Germany). Specific primers were designed using Primer 5.0 and listed in Table 1. Normal PCR was performed to analyze the expression of *Amh* and *Amhr2* in COCs, cumulus cells, and cumulus-free oocytes, and the PCR products were run on 1.2% agarose (Biowest, Nuaillé, France) gel, and stained with GelRed (Vazyme, Nanjing, China). To access the mRNA expression of *Fshr*, *Amhr2*, *Bmp15*, *Gdf9*, *Ptgs2*, *Has2*, *Ptx3*, and *Tnfaip6* genes in COCs, quantitative reverse-transcription PCR (RT-qPCR) was performed under the guide of instruction of Quantinova SYBR Green PCR Kit (QIAGEN, Dusseldorf, Germany) by CFX96 real-time PCR detection system (Bio-Rad, Hercules, CA, USA). The reaction conditions were as follows: 95 °C for 1 min, 40 cycles of amplifications (95 °C for 10 s, 60 °C for 30 s, and 72 °C for 15 s). Melting curve analysis was performed in the range of 65 °C to 95 °C, 0.5 °C per 5 s increments. Each sample was run along with a no-template control (NTC). The amplification efficiency of all primers was between 90 and 110%. The relative expression of genes was calculated by 2^−ΔΔCT^ [33] and *β-actin* was used as internal reference gene.

### 2.5. Detection of MPF and cAMP Contents in Oocytes

After culturing COCs in droplets with different treatments for 16 h, COCs were transferred into 200 μL of α-MEM with 0.1% hyaluronidase (Sigma-Aldrich, St. Louis, MO, USA). After instant centrifugation, about 40 oocytes were picked up with a thin glass tube and transferred into 20 μL of PBS with pH of 7.2–7.4. The samples were destroyed at −80 °C, and supernatants were collected after centrifugation at 2500 rpm for 20 min. Maturation-promoting factor (MPF) and cyclic adenosine 3’, 5’-monophosphate (cAMP) contents were detected according to the manufacturer’s instructions (Mlbio, Shanghai, China). The intra-assay and inter-assay coefficients of variation in MPF and cAMP were less than 10% and 15%, respectively.

### 2.6. Evaluation of Cumulus Expansion

The ability of AMH to regulate cumulus expansion was analyzed by adding 100 ng/mL rhAMH and 100 ng/mL rmFSH alone or in combination to the COCs in vitro culture system. After 16 h culturing of COCs, the cumulus expansion index (CEI) was calculated according to the previously reported method [34]. Briefly, cumulus expansion can be divided into five levels: grade 0, no cumulus expansion, oocytes attached to the bottom of the dish; grade 1, only the outermost 1–2 cumulus granulosa cells expanded; grade 2, the outer cumulus granulosa cells expanded radially, and the whole COCs were observed to be fluffy; grade 3, the radial crown part did not expand, the rest were expanded; grade 4, all cumulus granulosa cells expanded. CEI = [(number of grade 0 oocytes × 0) + (number of grade 1 oocytes × 1) + (number of grade 2 oocytes × 2) + (number of grade 3 oocytes × 3) + (number of grade 4 oocytes × 4)]/total number of oocytes.

### 2.7. Measurement of Estrogen and Progesterone

After 16 h culturing of COCs, culture supernatant was collected for hormone detection. Estradiol and progesterone were measured using the mouse estradiol (E2) ELISA kit (CUSABIO, Wuhan, China) and mouse progesterone (PROG) ELISA kit (CUSABIO, Wuhan, China) according to the instructions. The intra- and inter-assay coefficients of variation were less than 15.0% and 15.0% for estradiol, 15.0% and 15.0% for progesterone, respectively.

### 2.8. Statistical Analysis

All data were presented as mean ± SEM (standard error of mean) and each experiment was conducted at least in triplicate. Cumulus expansion index (CEI) was analyzed by SPSS Kruskal–Wallis test followed by Holm adjustment, and other data analysis was performed using SPSS software package and one-way ANOVA followed by least significant difference (LSD) test. Differences were considered to be statistically significant when *p* < 0.05.

## 3. Results

### 3.1. Expression of Amh and Amhr2 in COCs, CCs and Cumulus-Free Oocytes

As shown in Figure 1, *Amh* and *Amhr2* transcripts were detected in both mouse COCs and CCs. In contrast, *Amhr2*, but not *Amh* transcripts, was observed in cumulus-free oocytes (Figure 1A,B, Lane 3), suggesting AMH may exert paracrine effects on oocytes maturation through binding to AMHR2.

### 3.2. Effect of AMH on In Vitro Maturation of Cumulus Oocyte Complexes

The effect of AMH on COCs in vitro maturation was analyzed by treatment with 100 ng/mL rhAMH and 100 ng/mL rmFSH alone or in combination. As shown in Figure 2A, treatment with FSH resulted in the highest oocyte maturation rate with 93% reaching MII. Treatment with 100 ng/mL rhAMH had no significant effect on the COCs maturation when compared to the control group. The maturation rate of COCs in the combination of AMH and FSH group was decreased (*p* < 0.05) when compared to FSH alone treatment group, suggesting that AMH could inhibit the promoting effect of FSH on COCs in vitro maturation. Likewise, FSH induced higher mRNA expression of *Fshr* (Figure 2B, *p* < 0.05), AMH was able to attenuate this effect although there was no significant difference between the FSH alone and FSH plus AMH groups. The expression of *Amhr2* transcript was upregulated in all experimental groups (Figure 2C, *p* < 0.05). Unexpectedly, there were no differences among groups in the mRNA expression of either *Bmp15* or *Gdf9* genes (Figure 2D,E). Compared with the AMH alone treatment group, the AMH and FSH combined groups could reduce the cAMP content (Figure 2F, *p* < 0.05), while the AMH and FSH alone treatment group had no significant difference in the content of cAMP as compared to the control group. In addition, the FSH treatment group increased the MPF content of oocytes (Figure 2G, *p* < 0.05), whereas the AMH and FSH combined treatment group resulted in a significant reduction in the MPF content stimulated by FSH (Figure 2G, *p* < 0.05).

### 3.3. Effect of AMH on Cumulus Expansion of Cumulus Oocyte Complexes

The results showed that the AMH treatment group had no effect on cumulus expansion (Figure 3A–C), while FSH increased cumulus expansion (Figure 3B,C, *p* < 0.05). Compared with the FSH alone, the combination of AMH and FSH resulted in a decrease in the cumulus expansion index (Figure 3B,C, *p* < 0.05), suggesting that AMH may inhibit the stimulatory effect of FSH on cumulus expansion. RT-qPCR was used to further detect transcripts associated with cumulus expansion in each treatment group. As shown in Figure 4, AMH treatment increased the mRNA expression of *Ptgs2* in cumulus cells, whereas the expressions of *Has2*, *Ptx3,* and *Tnfaip6* genes were unchanged (*p* > 0.05). In contrast, FSH upregulated the mRNA expressions of *Has2*, *Ptx3,* and *Tnfaip6* transcripts (*p* < 0.05). Furthermore, AMH inhibited the stimulatory effects of FSH on *Has2*, *Ptx3,* and *Tnfaip6* expressions.

### 3.4. The Role of AMH in Regulation of Estradiol and Progesterone

After 16 h culturing of COCs, estradiol and progesterone in culture supernatant were measured to detect the effect of AMH on steroidogenesis in COCs. The results showed that there were no significant differences in estradiol content among AMH, FSH, AMH plus FSH and the control groups (Figure 5A, *p* > 0.05). Compared with the control group, there was no significant difference in progesterone content in the AMH alone treatment group (Figure 5B, *p* > 0.05), while the FSH treatment group could significantly increase progesterone production (Figure 5B, *p* < 0.05). Importantly, AMH significantly inhibited the promoting effect of FSH on progesterone levels (Figure 5B, *p*< 0.05).

## 4. Discussion

Several studies have shown that AMH and its receptor are mainly expressed in granulosa cells of non-atretic, preantral, and small antral follicles [20,35]. In the present study, we investigated the expression levels of *Amh* and *Amhr2* in cumulus cells and cumulus oocyte complexes (COCs) as well as cumulus-free oocytes. We found that *Amh* is exclusively expressed in murine cumulus cells. In contrast, *Amhr2* is expressed in both oocytes and cumulus cells. These results are in agreement with a recent study [32], although previous studies reported that oocytes expressed very little amount or no *Amh* and *Amhr2* mRNA [20]. On the other hand, the present results confirmed that *Amh* remains highly expressed in cumulus cells, which is supported by previous reports in humans, indicating AMH is predominantly expressed in cumulus cells of large antral and pre-ovulatory follicles [29,36].

Considering our observations that *Amhr2* expressed in oocytes, in the current study we investigated whether AMH influences the in vitro maturation of cumulus oocyte complexes (COCs). Notably, contradictory results have been reported concerning the direct effects of AMH on COCs maturation. An early study on the actions of AMH in the ovary indicated that bovine AMH inhibited oocyte meiosis in rats [30], whereas other studies indicated that AMH had no effect on oocyte meiosis in rats [31] and mice [32]. Our result confirmed that AMH has no significant effects on in vitro COCs maturation. It is generally appreciated that FSH supplementation in the maturation medium can enhance the in vitro oocyte maturation [37,38]. In this study, we observed that FSH of 100 ng/mL concentrations improved the COCs quality compared to the control group. Interestingly, we also found that AMH suppressed the nuclear maturation of COCs induced by FSH, accompanied by reductions in MPF content and *Fshr* expression. There are many reports clearly showing that AMH exerts an inhibitory role in follicular sensitivity to FSH [24,25,39] and FSH receptor expression [39]. Our findings here further broaden this negative effect of AMH on FSH inducing COCs in vitro maturation.

Cumulus expansion of the cumulus oocyte complexes (COCs) is necessary for meiotic maturation of oocytes, and regulated by endocrine and paracrine factors including FSH [38,40], epidermal growth factor (EGF) [41,42], and Insulin-like growth factor 1 (IGF-1) [43]. In this study, when COCs were treated with the combination of AMH and FSH, the stimulatory effect of FSH was significantly inhibited, which is consistent with the COCs maturation results. *Ptgs2*, *Has2*, *Ptx3,* and *Tnfaip6* are the key genes involved in the cumulus expansion; therefore, we further determined the mRNA expression of those genes. Similarly to recently published results [38], we found that the stimulatory effect of FSH on cumulus expansion was associated with a marked upregulation of *Has2, Ptx3,* and *Tnfaip6* expression, responses believed to be mediated mainly through protein kinase A (PKA) and EGF pathways [41] as well as an estrogen-signaling pathway mediated by G-protein coupled receptor 30 (GPR30) [38]. However, AMH can block the FSH-stimulating effect on mRNA expression of *Has2, Ptx3,* and *Tnfaip6*. Those results demonstrated that AMH has a negative regulatory effect on FSH biological function through limiting cumulus expansion.

Some reports demonstrated that AMH has a negative and inhibitory role on FSH-stimulated estradiol production in human granulosa-lutein cells by decreasing FSH-stimulated aromatase expression [26,27]. Here, we observed that AMH had no effect on basal and FSH-induced estradiol levels in the supernatant of COCs culture medium. The discrepant observations of AMH on estradiol production may be due to different culture materials. Notably, we found that AMH reduced FSH-stimulated progesterone production, which is similar to the previous report showing that AMH inhibited EGF-stimulated progesterone production in human granulosa-luteal cells [44].

## 5. Conclusions

The results of this study indicate that AMH has no effect on COCs nuclear maturation and cumulus expansion. Furthermore, AMH has an inhibitory effect on FSH-stimulated COCs maturation, cumulus expansion, and progesterone production. Our findings further broaden the horizon for our understanding of the actions of AMH on the oocyte and the inhibitory effects of AMH on the biological activity of FSH during follicular development.

## Figures and Tables

**Figure 1 animals-12-01209-f001:**
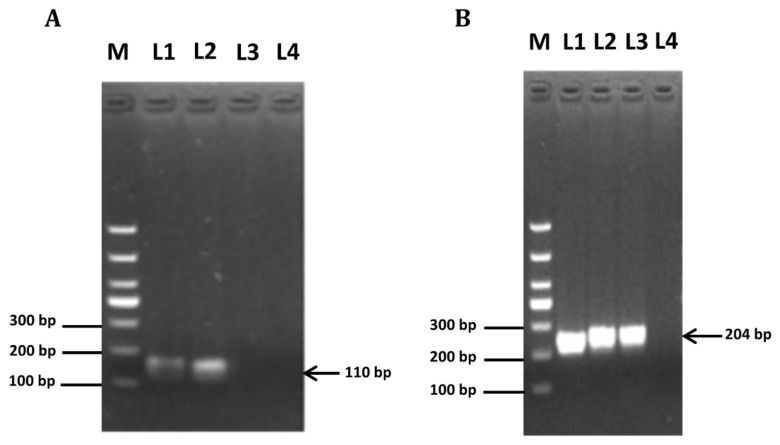
Expression of *Amh* and *Amhr* in mouse cumulus cells and oocytes. PCR amplification product of representative agarose gel electrophoresis. RNA was isolated from cumulus cells (Lane 1), cumulus oocyte complexes (Lane 2), and cumulus-free oocytes (Lane 3). One-microliter amounts of cDNA were used as templates for *Amh* (**A**) and *Amhr* (**B**) amplification, ddH_2_O was used as negative control for PCR amplification (Lane 4).

**Figure 2 animals-12-01209-f002:**
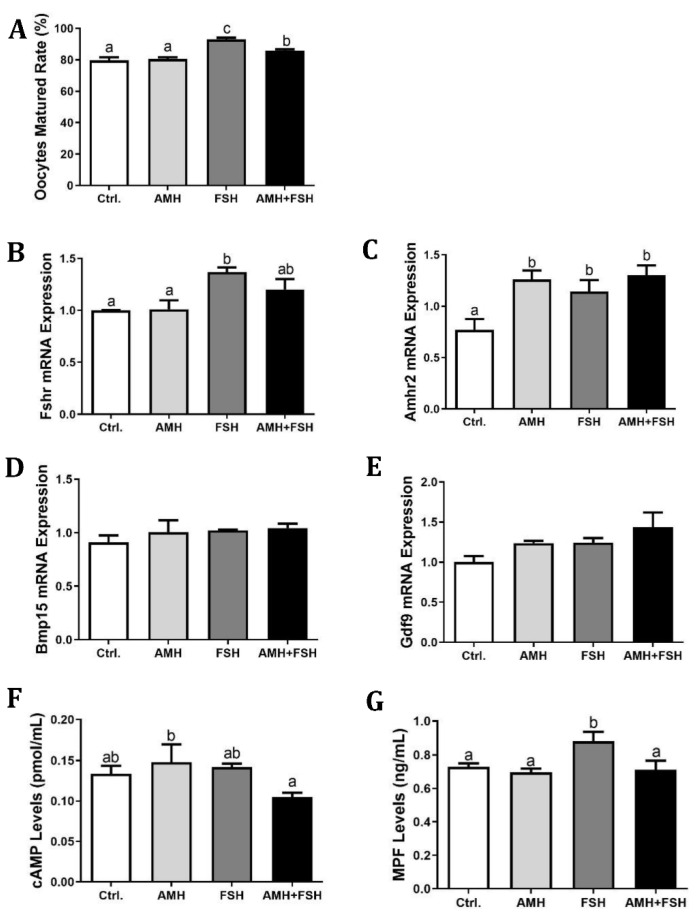
Effect of rhAMH on COCs in vitro maturation. COCs were supplemented with AMH and FSH alone or in combination for 16 h, the first polar body extrusion, mRNA expression in cell lysate, cAMP and MPF levels in the cytoplasm were analyzed. (**A**), Polar body extrusion percentage; (**B**–**E**), the relative mRNA expression of *Fshr*, *Amhr2, Bmp15*, and *Gdf9*; (**F**), cAMP levels; (**G**), MPF levels. Data were expressed as mean ± SEM from at least three independent experiments. Bars with different letters represent significant differences (*p* < 0.05).

**Figure 3 animals-12-01209-f003:**
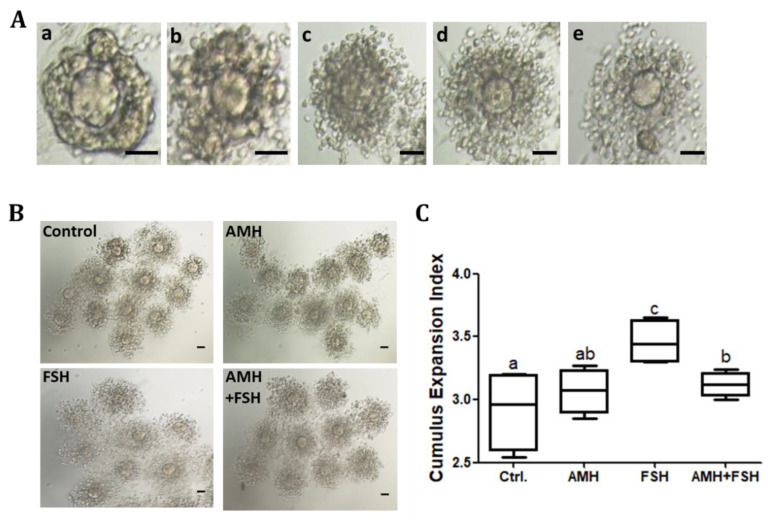
Effect of rhAMH on cumulus cell expansion. COCs were supplemented with AMH and FSH alone or in combination for 16 h, cumulus cell expansion was analyzed by the cumulus expansion index (CEI). (**A**), representative images of different grades of cumulus cell expansion, a = 0, b = 1, c = 2, d = 3, e = 4, the scale bar represents 50 μm. (**B**), the representative cumulus cell expansion of each group, the scale bar represents 50 μm. (**C**), the CEI of each group. Data were from at least three independent experiments. Bars with different letters represent significant differences (*p* < 0.05).

**Figure 4 animals-12-01209-f004:**
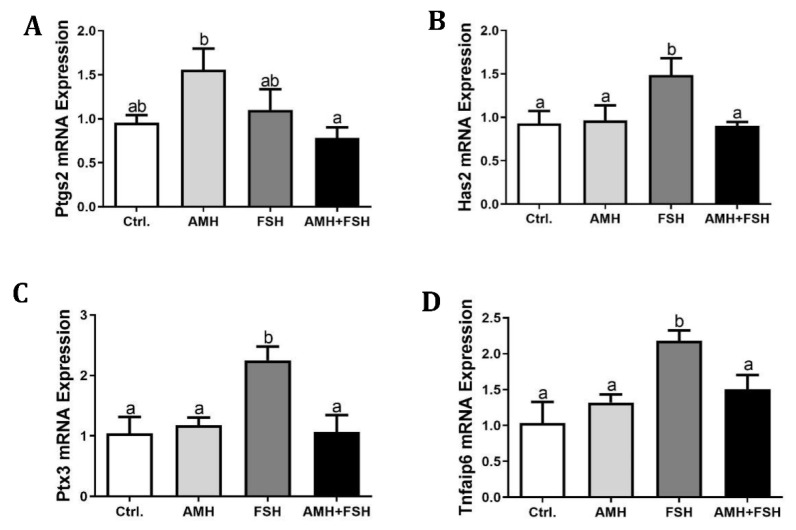
Effect of rhAMH on expression of cumulus expansion related genes. COCs were supplemented with AMH and FSH alone or in combination for 16 h,relative expression of transcripts for *Ptgs2* (**A**), *Has2* (**B**), *Ptx3* (**C**), and *Tnfaip6* (**D**) genes in cumulus cells were detected by RT-qPCR. The results were evaluated as the relative ratio of the expression level of each mRNA to that of *β-actin* and were expressed as mean ± SEM from three independent experiments. Bars with different letters represent significant differences (*p* < 0.05).

**Figure 5 animals-12-01209-f005:**
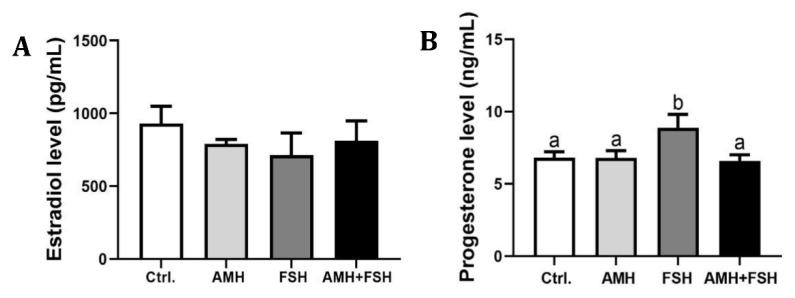
Effect of rhAMH treatment on estradiol (**A**) and progesterone (**B**) productions in COCs. The data were expressed as mean ± SEM from three independent experiments. Bars with different letters represent significant differences (*p* < 0.05).

**Table 1 animals-12-01209-t001:** Primer information in this study.

Gene	Primer	Sequence 5′–3′	Product Length (bp)	Accession No.
*Amh*	Forward	TACTCGGGACACCCGCTATT	110	NM_007445.3
	Reverse	TCAGGGTGGCACCTTCTCT		
*Amhr2*	Forward	GCAGCACAAGTATCCCCAAAC	204	NM_001356575.1
	Reverse	GTCTCGGCATCCTTGCATCTC		
*Fshr*	Forward	AGGTACAGCTCTGCCATGCT	171	NM_013523.3
	Reverse	GTACGAGGAGGGCCATAACA		
*Gdf9*	Forward	TGGAACACTTGCTCAAATCGG	106	XM_006532220.5
	Reverse	GACATGGCCTCCTTTACCACA		
*Bmp15*	Forward	GAAAATGGTGAGGCTGGTAAAG	153	NM_009757.5
	Reverse	AGATGAAGTTGATGGCGGTAAA		
*Ptx3*	Forward	TTTGGAAGCGTGCATCCTGT	186	NM_008987.3
	Reverse	GTTCTCCTTTCCACCCACCA		
*Ptgs2*	Forward	GTTCATCCCTGACCCCCAAG	193	NM_011198.4
	Reverse	TCCATCCTTGAAAAGGCGCA		
*Tnfaip6*	Forward	GCTCAACAGGAGTGAGCGAT	166	NM_009398.2
	Reverse	CTGACCGTACTTGAGCCGAA		
*Has2*	Forward	GACGACAGGCACCTTACCAA	116	NM_008216.3
	Reverse	TGCTGGTTCAGCCATCTCAG		
*β-actin*	Forward	TAAAGACCTCTATGCCAACACAGT	241	NM_007393.5
	Reverse	CACGATGGAGGGGCCGGACTCATC		

*Amh*: Anti-Müllerian hormone, *Amhr2*: Anti-Müllerian hormone receptor type 2, *Fshr*: Follicle-stimulating hormone receptor, *Gdf9*: Growth differentiation factor 9, *Bmp15*: Bone morphogenetic protein 15, *Ptx3*: Pentraxin 3, *Ptgs2*: Prostaglandin-endoperoxide synthase 2, *Tnfaip6*: TNF-alpha-induced protein 6, *Has2*: Hyaluronan synthase 2.

## Data Availability

The data that support the findings of this article are available from corresponding author upon reasonable request.

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
