# Peer review of "Anti-Müllerian Hormone Inhibits FSH-Induced Cumulus Oocyte Complex In Vitro Maturation and Cumulus Expansion in Mice"

_animals, 2022, doi:10.3390/ani12091209_

Round 1

Reviewer 1 Report

In this new version of the MS, the authors met most of the concerns raised during the revision. However, the MS may still be improved in some (minor) aspects:

  • the authors state, at the end of the abstract and at the end of the MS "Taken together, our results suggest that AMH may inhibit the effects of FSH-induced in vitro COC maturation and cumulus expansion. These findings increase our knowledge of the functional role of AMH in regulating folliculogenesis." But this is relatively wider information, that does not elucidate to the reader how this study clarified the role of AMH on folliculogenesis regulation. Could you please clarify/particularize it?
  • Please provide the necessary information regarding the controls used in ALL the steps of RT-PCR. the images' captions provide some insights about the control for the final step, but not for the initial ones.
  • In figure 3, The images in A.a & A.b seem to be obtained with a different lens than the other images. Can you confirm the magnification used? Can you provide images gathered using the same objective?
  • Also in figure 3, the photomicrographs are laterally flattened to become square. Please avoid distorting the images. Try to find a balance between the side of the microphotographs plate and the graph. if necessary, reduce the lateral size of the later

Other minor suggestions have been introduced in the commented copy of the MS. Finally, change "agricultural animals" (the name is not used) into livestock

Author Response

In this new version of the MS, the authors met most of the concerns raised during the revision. However, the MS may still be improved in some (minor) aspects:

Comment 1. The authors state, at the end of the abstract and at the end of the MS "Taken together, our results suggest that AMH may inhibit the effects of FSH-induced in vitro COC maturation and cumulus expansion. These findings increase our knowledge of the functional role of AMH in regulating folliculogenesis." But this is relatively wider information, that does not elucidate to the reader how this study clarified the role of AMH on folliculogenesis regulation. Could you please clarify/particularize it?

Response: Thanks very much for your comment. For precise description, we have changed this sentence as “These findings increase our knowledge of the functional role of AMH in regulating oocyte maturation.”  

Comment 2. Please provide the necessary information regarding the controls used in ALL the steps of RT-PCR. The images' captions provide some insights about the control for the final step, but not for the initial ones.

Response: Thanks for your kind reminder. We have added the sentence of “Each sample was run along with a no-template control (NTC)” in the Materials and Methods section (2.4) in the revised manuscript. 

Comment 3. In figure 3, the images in A.a & A.b seem to be obtained with a different lens than the other images. Can you confirm the magnification used? Can you provide images gathered using the same objective?

Response: Thanks very much for your careful review. According to your suggestion, we have checked the scale bar again and corrected the scale bar in the revised manuscript.

Comment 4. Also in figure 3, the photomicrographs are laterally flattened to become square. Please avoid distorting the images. Try to find a balance between the side of the microphotographs plate and the graph. if necessary, reduce the lateral size of the later.

Response: Thanks very much for your careful review. According to your suggestion, we have adjusted Figure 3B to avoid distorting them in the revised manuscript.  

Comment 5. Other minor suggestions have been introduced in the commented copy of the MS. Finally, change "agricultural animals" (the name is not used) into livestock.

Response: Thanks for your kind comments, we have already corrected them in the revised manuscript.

Reviewer 2 Report

I reviewed your manuscript. I think this manuscript contents are enough to published as Animals. thank you.

Author Response

Thanks very much for your review.

This manuscript is a resubmission of an earlier submission. The following is a list of the peer review reports and author responses from that submission.

Round 1

Reviewer 1 Report

In their MS, Xinzhe Zhao and colleagues describe their findings on the treatment of cultured mice oocytes (naked, COC, and GC) in an in vitro maturation system with recombinant human AMH. Treatment results were evaluated by the synthesis and protein of multiple molecules, the MPF and cAMP content in oocytes, and the evaluation of cumulus expansion and the 1st globar pole extrusion. The authors also established the synthesis of AMH and AMHR2 in non-treated structures.

The study generated many data that may bring some additional information on the AMH role in oocyte maturation.

However, the information provided in the paper is insufficient in some aspects (particularly in Material & Methods). Authors should provide additional information regarding:

  • the exact number of samples used in each group and for each technique/assay - we are aware that 120 oocytes were used in total, but nothing else.
  • please describe the controls used for the Rt-PCR technique. Also, mention how the mRNA values were estimated
  • please provide the complete identification (reference and clone) of the primary antibodies used in WB
  • authors should explain why did they use two different control molecules (b-actin for RT-PCR and GAPDH for WB)...
  • this fact may bias the results, as the comparisons may be somehow skewed by the reference molecule being different
  • In the results section, the authors refer to using the ELISA method to quantify the MPF and cAMP content. Such information should be provided in M&M, together with the intra-assay variation (and inter-assay also of multiple runs/plates were used)
  • Also, in the results section, the authors report data for oocyte/COCs treatments with rhAMH and FSH, but no information about those treatments is provided in M&M - please add it with the necessary detail
  • in statistical analysis: please add information regarding the exploitation of the normal distribution of data that supports the use of parametric tests. Also, in the discussion, the authors refer to a correlation between oocyte maturation and AMH treatments, but no information was provided in M&M about the use of correlation tests.
  • Cumulus expansion index is a non-numeric variable. The Authors should explain which test they used to estimate the groups´ differences (as presented in Figure 5). A chi-square analysis should test comparisons between the groups (frequency of distribution of the index within groups)
  • in M&M, it is stated that GAPDH was the reference molecule used in WB, but in some graphs in results, b-actin was identified as the control molecule. Please correct which information is at fault
  • describe how the value for Rt-PCR and WB was estimated
  • provide information regarding the estimation of the cumulus expansion index

IN the results section, there is a short amount of information that belongs to M&M. please provided the needed detail and move it into that section.

Figure 5 - the photomicrographs are laterally flattened (in different degrees between images). Correct and retain the natural proportion in the original images. Also, photographs should (all of them) present the scale bar.

In the discussion section, authors compare controversial data from different studies (lines 305-fw.). I want to challenge the authors to interpret and draw inferences from the different studies in the light that the homology of AMH molecule varies between species. So the species of origin is a piece of important information when comparing different studies.

Finally, in all the figure captions, the authors should provide information on the number of samples/cells or structures used in each group and test.

Some additional minor comments were included in the commented copy of this MS attached to this review.

Reviewer 2 Report

General comments:

The manuscript entitled “AMH promotes maturation of naked oocytes but inhibits FSH-induced COC maturation and cumulus expansion” (Manuscript ID; animals-1233596) was describes effects of Anti-Mullerian Hormone (AMH) on nuclear maturation of naked oocytes, nuclear maturation and cumulus expansion of cumulus oocyte complex (COC) in in vitro culture system. From the results, AMH increased the maturation rate of naked oocyte caused by through AMHR2, however, AMH did not increase the maturation rate of COC. In addition, although FSH increased the maturation rate of COC, FSH+AMH did not increase maturation rate of COC. The caused of inhibition of oocyte maturation on COC by FSH+AMH were caused by inhibition of cumulus expansion. The manuscript is very interesting to inform about the role of AMH on oocyte maturation. There are some points needing corrections in the manuscript. Please consider the suggested edits listed below.    

Through the manuscript

In your manuscript, the name of gene converted to Italic. However, some words did not convert to Italic. Please check and unify the format of the name of gene to Italic

P2 L75-76

Please describe the species of this reference.

P3 L105

Please space between “with” and “100”.

P3 L125

Please space between “,” and “California”.

P3 L126

Please space between “,” and “Specific”.

P3 L131

Please space between “,” and “Spain”

P5 L173

I think the data of cumulus expansion index (CEI) is categorical data. Therefore, you cannot analysis by ANOVA, you must use Kruskal-Wallis test followed by Holm’s adjustment for multiple comparison test or Steel-Dwass test. Also, because the data of CEI is categorical data, you must show the Figure 5 C by box plot.

P11 Figure6 A

Please change “ptgs2” to “Ptgs2”

P12 L

Please more discuss about the reason of the inhibition of AMH+FSH group.

You showed the results of expression of Ptgs2, Has2, Ptx3, and Tnfair6 among 4 groups. However, I could not well understand the relation between mRNA expression differences and the inhibition of maturation rate in AMH+FSH compared to FSH treatment group. Please describe more precisely in the Discussion.

 In addition, if you want to discuss about the reason of the inhibition of oocyte maturation in AMH+FSH caused by lower effect of FSH on cumulus expansion, I think you had to analysis the expression of FSHR in only cumulus cell.  By doing so, the cause of inhibition of Has2, Ptx3, and Tnfair6 expressions in AMH+FSH were clarified whether the downregulation of FSHR caused the inhibition of oocyte maturation rate.

Round 2

Reviewer 1 Report

in their revised MS, the authors addressed most of my comments. I would like to invite them to identify that the use of different control genes in WB and RT-PCR assays may rend difficult a direct transposition/comparison of the data. This issue can be viewed as a minor shortcome in this paper and should be clearly presented

Some minor comments were included in the commented copy of the MS that is attached to this review

Author Response

Q1: In their revised MS, the authors addressed most of my comments. I would like to invite them to identify that the use of different control genes in WB and RT-PCR assays may rend difficult a direct transposition/comparison of the data. This issue can be viewed as a minor shortcome in this paper and should be clearly presented.

A1: Thanks for your rigorous comments. It’s really better to perform the same internal gene in both WB and RT-PCR. Actually, both β-actin and GAPDH are two widely used housekeeping genes to normalize the variability in template loading due to their stable expression in oocyte. Sometimes, researcher choose the different housekeeping gene in the manuscript depending on the molecular size of target protein. To resolve this minor shortcome, we decided to re-perform WB using β-actin as control to keep in line with RT-PCR which used β-actin as internal gene. However, it need more time to re-perform because we need prepare the samples and purchase antibodies from overseas, and that will take 4-6 weeks. We hope our next results will meet your suggestions.    

Q2: Some minor comments were included in the commented copy of the MS that is attached to this review

A2: Thanks very much for your careful revision. We have corrected them in the revised manuscript. By the way, we also checked the primer information carefully and make a minor revision.
